# Long-term efficacy of mandibular advancement devices in the treatment of adult obstructive sleep apnea: A systematic review and meta-analysis

Min Yu[1,2,3], Yanyan Ma[4], Fang Han[5☯]*, Xuemei Gao [1,2,3☯]*

1 Department of Orthodontics, Peking University School and Hospital of Stomatology, Beijing, P.R. China,
2 Center for Oral Therapy of Sleep Apnea, Peking University Hospital of Stomatology, Beijing, Haidian District, P.R. China, 3 National Center for Stomatology, Beijing, Haidian District, P.R. China, 4 Department of Stomatology, Beijing Chao-Yang Hospital, Capital Medical University, Beijing, Chaoyang District, P.R. China, 5 Sleep Division, Peking University People's Hospital, Beijing, Xicheng District, P.R. China

☯ These authors contributed equally to this work.
* hanfang1@hotmail.com (FH); xmgao@263.net (XG)

**Data Availability Statement:** All relevant data are within the paper and its Supporting Information files.

## Abstract

This study aims to review the long-term subjective and objective efficacy of mandibular advancement devices (MAD) in the treatment of adult obstructive sleep apnea (OSA). Electronic databases such as PubMed, Embase, and Cochrane Library were searched. Randomized controlled trials (RCTs) and non-randomized self-controlled trials with a treatment duration of at least 1 year with MAD were included. The quality assessment and data extraction of the included studies were conducted in the meta-analysis. A total of 22 studies were included in this study, of which 20 (546 patients) were included in the meta-analysis. All the studies had some shortcomings, such as small sample sizes, unbalanced sex, and high dropout rates. The results suggested that long-term treatment of MAD can significantly reduce the Epworth sleepiness scale (ESS) by -3.99 (95%CI -5.93 to -2.04, $p<0.0001$, $I^2 =$ 84%), and the apnea-hypopnea index (AHI) -16.77 (95%CI -20.80 to -12.74) events/h ($p<0.00001$, $I^2 = 97\%$). The efficacy remained statistically different in the severity (AHI<30 or >30 events/h) and treatment duration (duration <5y or >5y) subgroups. Long-term use of MAD could also significantly decrease blood pressure and improve the score of functional outcomes of sleep questionnaire (FOSQ). Moderate evidence suggested that the subjective and objective effect of MAD on adult OSA has long-term stability. Limited evidence suggests long-term use of MAD might improve comorbidities and healthcare. In clinical practice, regular follow-up is recommended.

## Introduction

Obstructive sleep apnea (OSA) is a complicated chronic condition, which has emerged as a very relevant public health problem because of its high prevalence [1]. In addition to low quality of life, patients with OSA often suffer from unrefreshing sleep, daytime fatigue, memory loss, and even long-term effects such as cardiovascular, metabolic, cognitive, and cancer-

**Funding:** This study was supported by the Clinical Research Foundation of Peking University School and Hospital of Stomatology (PKUSS-2023CRF106) granted to M.Y. The funders had no role in study design, data collection and analysis, decision to publish, or preparation of the manuscript.

**Competing interests:** The authors have declared that no competing interests exist.

related alterations [2]. OSA is characterized by complete (apnea) or partial (hypopnea) cessation of airflow during sleep, causing oxygen desaturation and fragmentation of sleep [3]. Various treatment options have been used to treat patients with OSA, including behavioral modifications, such as weight loss and alcohol avoidance [4]; non-surgical interventions, such as continuous positive airway pressure (CPAP) and oral appliances (OA); and surgeries, such as uvulopalatopharyngoplasty (UPPP) and maxillomandibular advancement (MMA). Mandibular advancement devices, designed to advance the mandible, are the most commonly used oral appliances [5], indicated for use in patients with mild to moderate OSA and those who do not tolerate nor prefer CPAP by the American Academy of Sleep Medicine (AASM) [6].

Many randomized controlled trials (RCTs) with high-quality evidence have confirmed that MAD can effectively reduce respiratory events during sleep, improving daytime sleepiness and quality of life, whereas the efficacy varied among patients [7–18]. MADs act by shifting the mandible forward, which could keep the mandible from backward rotation, widen the lateral dimension of the upper airway, tension the soft palate and stabilize the hyoid bone during sleep [19–21]; however, long-term use of MAD has been found to cause bite change, which might influence the efficacy [22]. OSA requires lifelong treatment, and the long-term efficacy of MAD on OSA is important for clinical decision-making. With the increasing number of relevant research publications in recent years, this study intends to systematically summarize the long-term (treatment time > 1 year) subjective and objective efficacy of MAD in the treatment of adult OSA.

## Materials and methods

The Preferred Reporting Items for Systematic Reviews and Meta-Analyses (PRISMA) statement checklist was followed in the study [23].

### Inclusion criteria

The inclusion criteria were based on the elements of PICO (patient, intervention, comparison, and outcome).

Population: Adult patients diagnosed with OSA.

Intervention: Mandibular advancement oral appliance treatment with a duration of at least one year.

Comparison: Post- and pre-treatment self-controlled comparisons.

Outcome: Measurable outcomes of efficacy, including objective parameters (apnea-hypopnea index; oxygen desaturation index, ODI; the lowest oxygen desaturation, $LSpO_2$, respiratory disturbance index, RDI, etc), and subjective parameters (Epworth sleepiness scale, ESS; Pittsburgh Sleep Quality Index, PSQI, etc).

### Search strategy

The systematic literature search was conducted in electronic databases on 30 June 2023 using pre-specified search terms, including PubMed (MEDLINE), EMBASE, and Cochrane Library, with keywords such as ('breathing, sleep disordered' OR 'obstructive sleep apnea') and ('mandibular advancement' OR 'oral appliance') and ('efficacy' AND 'long term'). Manual searches of the reference lists were completed for relevant studies.

### Study selection

Two reviewers (M. Yu and Y.Y. Ma) independently screened the titles and abstracts for potential eligibility. Any discrepancies were resolved by discussing each other and consulting with a

third reviewer (X.M. Gao). All the articles with retrieved full texts were read thoroughly. Conference abstracts, reviews, personal opinions, books, and articles not written in English were excluded.

### Data extraction

Two reviewers independently extracted the data (M. Yu and Y.Y. Ma). Study-specific data were collected, including the author, year, study design, the oral appliance's design, subjects' demographic characteristics, severity, treatment duration, and long-term efficacy. Outcomes of both post- and pre-treatment were recorded. If the mean (M) and standard deviation (SD) were not reported in the study, the estimated values were used in the meta-analysis [24, 25].

### Quality assessment

The risk of bias in included studies was assessed according to the risk of bias assessment tool for non-randomized studies (RoBANS) on six domains, including the selection of participants, confounding variables, measurement of exposure, blinding of outcome assessments, incomplete outcome data, and selective outcome reporting [26]. The selection of participants was rated high risk if the study was retrospective or patients were not consecutively recruited. The confounding variables mainly focused on the sex, age, severity, and systematic diseases of the participants. If the study lacks the time and frequency of the use of MAD, the measurement exposure would be rated high risk. Two reviewers independently evaluated the quality of the studies; a third reviewer (X.M. Gao) was consulted when there was disagreement.

### Data synthesis

A meta-analysis was conducted if there were enough high-quality studies. The data synthesis was performed using Review Manager 5.4 (The Cochrane Collaboration). The heterogeneity among studies was represented by the $I^2$ index and the $\chi^2$ test. Meta-analysis was performed with the fixed-effects model if $I^2 < 50\%$, otherwise the random-effects model would be implemented. Subgroup analyses were performed based on the duration of MADs treatment and initial severity. Funnel plots were used to assess the risk of publication bias.

## Results

### Search and study selection

Fig 1 illustrates the flowchart of the study selection process. A total of 502 articles were identified, 221 in PubMed, 212 in Embase, 69 in Cochrane Library, and two by manual search. There were 369 articles remaining after duplications were removed. Irrelevant articles were excluded after reading the title and abstract, and 23 articles were assessed with full text. Three articles were excluded because part of the subjects' treatment duration was less than a year. Finally, there were two articles added to the study by additional sources. The characteristics of the included studies are presented in Table 1. Most studies were before-after trials, and some studies were prospective long-term studies of short-term RCTs [27–35]. The longest treatment duration of MADs was more than 10 years [31, 36, 37]. Most studies included patients diagnosed with OSA, except the study conducted by de Godoy et al., which focused on the upper airway resistance syndrome (UARS) [38] and was excluded from data synthesis.

The majority of participants are middle-aged men, and participants suffered from mixed severity of OSA, from mild to severe, as defined by AHI. In all studies, there was polysomnographic confirmation of OSA and post-treatment efficacy was evaluated with

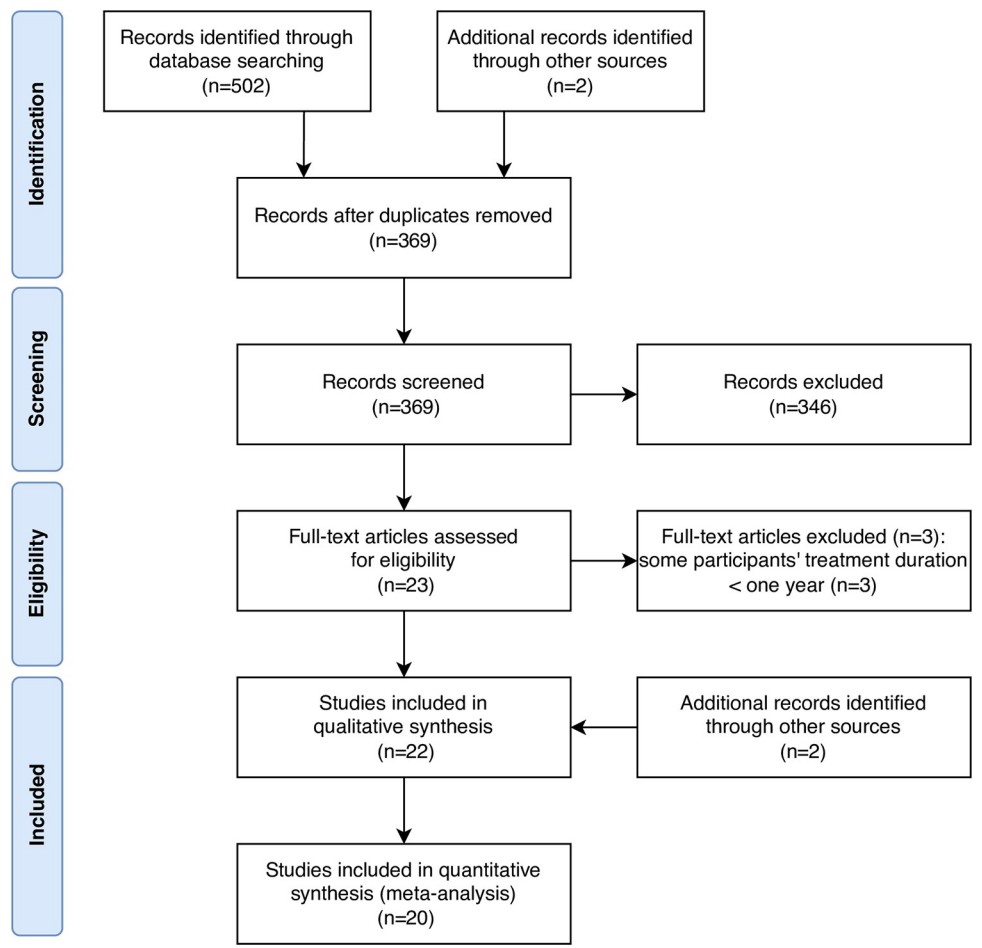

**Fig 1. The flowchart of the study selection process.**

polysomnography (PSG) or home sleep apnea test (HSAT). The dropout rate varied greatly across studies, ranging from 0% to 99%, with higher dropout rates in longer follow-up studies.

## Quality assessment

The results of the quality analysis of included studies are shown in Fig 2A and 2B. The selection of participants was rated high risk in most studies due to the unbalanced sex composition. The selective reporting bias was rated high in the study of Wilhelmsson et al. [27], due to the lack of detailed pre- and post-treatment results.

## Meta-analysis

The subjective efficacy represented by ESS and PSQI with long-term use of MAD in the treatment of OSA is shown in Fig 3A and 3B, respectively. There was a significant effect in favor of post-treatment in ESS, which decreased by -3.99 (95%CI -5.93 to -2.04, $p<0.0001$, $I^2 = 84\%$), whereas the change of PSQI score was not statistically different -0.59 (95%CI -1.70 to 0.52, $p = 0.30$).

The objective efficacy of the long-term use of MAD in treating respiratory events of OSA is shown in Fig 4A to 4F. The pooled analysis generated a heterogenous but significant result in AHI, with a decrease of -16.77 (95%CI -20.80 to -12.74) events/h ($p<0.00001$, $I^2 = 97\%$). The

**Table 1. Characteristics of the included studies.**

| Study | Study design | Treat-ment | Original sample (n) | Dropout (n, %) | Analyzed n (%) | M/F | Diagnosis | Age (y) | BMI (kg/m²) | Severity | Treatment duration | MAD design | Wearing time | Clinical examinations |
|---|---|---|---|---|---|---|---|---|---|---|---|---|---|---|
| Wilhelmsson et al., 1999 [27] | RCT* | MAD vs. UPPP | 49 | 12 (24%) | 37 (76%) | 37/0 | OSA | 49.3 (46.8 to 51.9) | 26.9 (25.6 to 28.3) | AHI 18.2 ±7.92# | 1y | 50% MMP, 5mm vertical opening | 6.2 nights/week | HSAT |
| Marklund et al., 2001 [39] | Prospective | MAD | 33 | 14 (42%) | 19 (58%) | 17/2 | OSA | 50±12 | 26±3.5 | AHI 22 ±17 | 5.2±0.4y | 5.3±1.4mm protrusion, 10±1.4mm vertical opening | 50 to 90% nights/week | PSG; questionnaire |
| Rose et al., 2002 [40] | Retrospective | Activator | 86 | 60 (70%) | 26 (30%) | 24/2 | OSA | 55.2 ±8.2 | 27.8 ±3.6 | AHI 17.8 ±8.5 | 1.5 to 2y | 1/2 to 3/4 the width of a premolar protrusion, 8 to 10mm vertical opening | >5h/night, daily | PSG; questionnaire |
| Fransson et al., 2003 [41] | Prospective | MAD | 50 | 6 (12%) | 44 (88%) | 38/6 | OSA | 56 (range 31 to 73) | 30 (range 21 to 38) | ODI 14.7 ±12.7 | 2y | - | 85% of the patients used every night | PSG; questionnaire |
| Tegelberg et al., 2003 [28] | RCT* | MAD | 74 | 19 (26%) | 55 (74%) | 55/0 | OSA | 50% 51.8 (49.0 to 54.6); 75% 54.4 (52.4 to 56.4) | 50% 27.4 (26.4 to 28.4); 75% 27.9 (26.6 to 29.3) | AHI 17.5 ±4.1 | 1y | 50%/75% MMP, 2mm vertical opening | - | HSAT; questionnaire |
| Itzhaki et al., 2007 [42] | Prospective | Herbst | 19 | 3 (16%) | 16 (84%) | 11/5 | OSA | 54.0 ±8.3 | 28.0 ±3.1 | ODI 5.9 ±9.9 | 1y | 75% MMP | - | PSG; HSAT; questionnaire |
| Ghazal et al., 2009 [29] | RCT* | MAD | 103 | 58 (56%) | 45 (44%) | 36/9 | OSA | 50.4 ±10.9# | 26.0 ±2.8# | AHI 34.5 ±7.5# | 2y | 83% MMP (5.5±2.7mm) | >5 nights/week | PSG; questionnaire |
| Aarab et al., 2011 [30] | RCT* | MAD vs. CPAP | 21 | 6 (29%) | 15 (71%) | - | OSA | 50.4 ±8.9 | 27.1 ±3.1 | AHI 21.4 ±11.0 | 1y | 25% MMP (n = 1), 50% MMP(n = 7), 75% MMP (n = 12) | - | PSG; questionnaire |
| Gauthier et al., 2011 [34] | RCT* | MAD | 16 | 2 (13%) | 14 (87%) | 10/4 | OSA | 51.9 ±1.7 | - | RDI 10.4 ±1.3 | Range 2.5 to 4.5 y | 81% to 96% MMP, 6.5 to 8.5mm vertical opening | 7.1 h/night, 6.4 nights/week | PSG; questionnaire |
| Doff et al., 2013 [35] | RCT* | MAD vs. CPAP | 51 | 22 (43%) | 29 (57%) | - | OSA | 49±10 | 32±6 | AHI 39 ±31 | 2.3±0.2 y | 50% MMP | 7.2±0.8 h/night; 6.7±0.7 nights/week | PSG; questionnaire |
| Gong et al., 2013 [43] | Retrospective | MAD | 412 | 318 (77%) | 94 (23%) | - | OSA | - | - | AHI 27.08 ±25.31# | 74 (30 to 99)m | 60 to 70% MMP, 4 to 5mm vertical opening | 6 to 8h/night, 5 to 7 nights/week | PSG; questionnaire; lateral cephalogram |

*(Continued)*

**Table 1.** (Continued)

| Study | Study design | Treat-ment | Original sample (n) | Dropout (n, %) | Analyzed n (%) | M/F | Diagnosis | Age (y) | BMI (kg/m²) | Severity | Treatment duration | MAD design | Wearing time | Clinical examinations |
|---|---|---|---|---|---|---|---|---|---|---|---|---|---|---|
| **Eriksson et al., 2014** [36] | Prospective | MAD | 77 | 32 (42%) | 45 (58%) | 35/10 | OSA or snoring | 54.0 ±8.0# | 29.3 ±3.8# | ODI 9.4 ±12.3# | 10y | - | - | PSG; questionnaire |
| **Ballanti et al., 2015** [44] | Prospective | MAD | 35 | 7 (20%) | 28 (80%) | 22/6 | OSA | 52.2 ±6.8 | 25.8 ±1.7 | AHI 12.4 ±3.6 | 2y | 75% MMP, 6mm vertical opening | - | PSG; questionnaire |
| **Marklund et al., 2016** [37] | Prospective | MAD | 630 | 621 (99%) | 9 (1%) | 8/1 | OSA | 51.7 (41.7, 59.1) | 26.5 (24.7, 31.1) | AHI 17.3 (9.7, 26.5) | 16.5 (16.3, 18.0) y | 6.0(5.0, 7.5) mm protrusion | - | PSG |
| **de Godoy et al., 2017** [38] | RCT | MAD vs. placebo | 36 | 6 (17%) | 30 (83%) | 21/9 | UARS | 43.7 ±7.7 | 26.6 ±4.1 | - | 1.5y | 50% MMP | 6.3±1.8h/night, 77% nights/week | PSG; questionnaire |
| **Gupta et al., 2017** [45] | Prospective | MAD | 30 | 0 (0%) | 30 (100%) | 25/5 | OSA | 41±4 | 22±5 | AHI 22 (5 to 30) | 2y | 70% MMP | - | PSG; blood pressure |
| **Knappe et al., 2017** [46] | Prospective | MAD | 43 | 29 (67%) | 14 (33%) | - | OSA | 54 | - | AHI 20.5(7 to 57) | 3y | 77.2% MMP | - | PSG; intraoral examinations |
| **Vigié du Cayla et al., 2019** [47] | Prospective | Somnodent®, ORM® | 24 | 0 (0%) | 24 (100%) | 15/9 | OSA | 54.3 ±12.6 | 27.2 ±5.7 | AHI 35.5 ±18.2 | 3.9±2.4y | 6.8(5 to 9) mm protrusion | - | PSG; questionnaire; lateral cephalogram |
| **Uniken Venema et al., 2020** [31] | RCT* | Thornton Adjustable Positioner | 51 | 37 (73%) | 14 (27%) | 12/2 | OSA | 61±8 | 32.4 ±6.6 | AHI 31.7 ±20.6 | 10.0±0.6y | 50%MMP | 7.8±0.9h/night, 6.6±1.0nights/week | PSG; questionnaire |
| **Baldini et al., 2022** [48] | Retrospective | AMO®, SomnoDent® | 444 | 327 (74%) | 117 (36%) | 81/36 | OSA | 62.0 (54.0, 69.0) | 26.0 (24.0, 28.0) | ODI 16.0(8.0, 27.0) | 4.6 (2.6, 6.6)y | 88.9(77.8, 100.0)% MMP | 7 h/night, 7 nights/week | HSAT; lateral cephalogram; questionnaire |
| **Lai et al., 2022** [32] | RCT* | MAD vs. CPAP | 52 | 5 (10%) | 47 (90%) | 44/3 | Severe OSA | 46.72 ±10.19 | 29.33 ±2.53# | AHI 33.55 ±6.52# | 1y | 4.5mm protrusion | Approximately 6 h/night, 5 nights/week | PSG; questionnaire; lateral cephalogram |
| **Luz et al., 2023** [33] | RCT* | MAD vs. CPAP vs. control | 25 | 6 (24%) | 19 (76%) | - | OSA | 44.8 ±15.1 | 28.2 ±7.2 | AHI 9.3 ±5.2 | 1y | The maximum comfortable protrusion | - | PSG; questionnaire |

*Randomized controlled trial (RCT) in a short-term period

# Calculated from the results of the study

BMI: body mass index; MAD: mandibular advancement device; UPPP: uvulopalatopharyngoplasty; OSA: obstructive sleep apnea; AHI: apnea-hypopnea index; MMP: maximal mandibular protrusion; HSAT: home sleep apnea test; PSG: polysomnography; ODI: oxygen desaturation index; CPAP: continuous positive airway pressure; UARS: upper airway resistance syndrome.

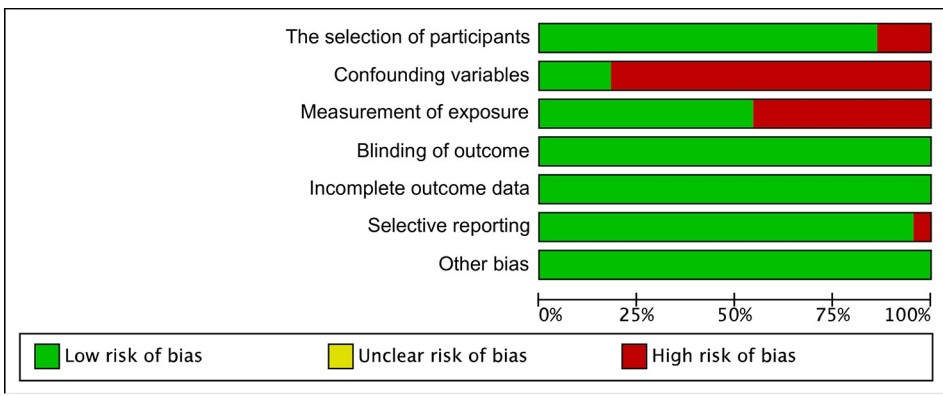

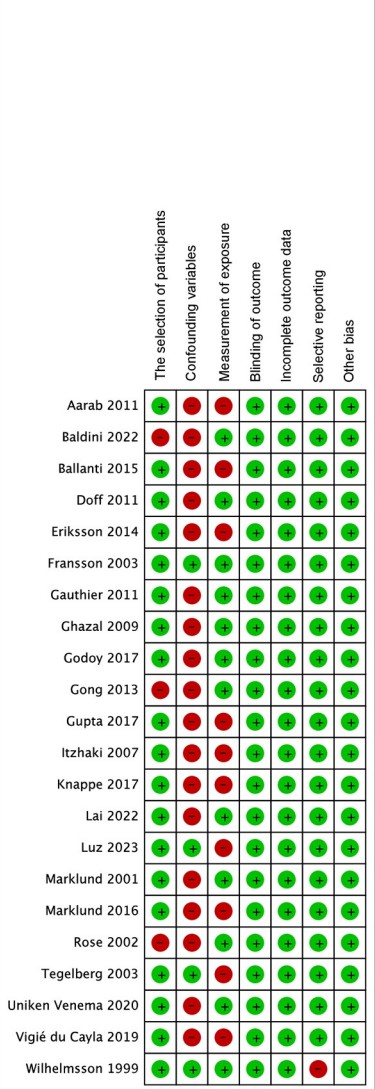

**Fig 2.** The summarized (a) and individual (b) risk of bias of included studies.

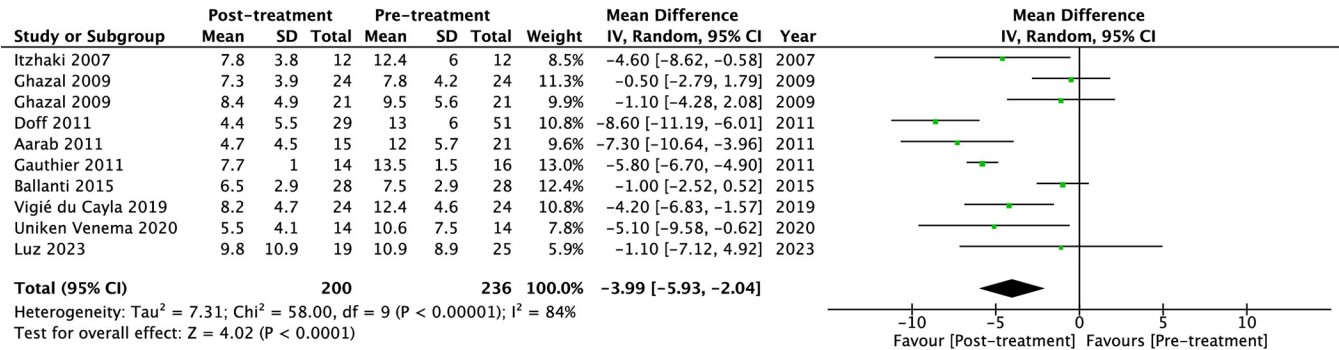

**Fig 3. The forest plot of the mean difference of subjective efficacy of mandibular advancement devices (MAD) on obstructive sleep apnea (OSA).** (a) Epworth sleepiness scale (ESS) and (b) Pittsburgh Sleep Quality Index (PSQI).

efficacy remained statistically different in the severity (AHI<30 or >30 events/h) and treatment duration (duration <5y or >5y) subgroups. As Fig 4C shows, the synthesized change of apnea index (AI) was -6.87 events/h (95%CI -8.08 to -5.66, $p$<0.00001, $I^2$ = 59%). ODI decreased by -16.93 events/h (95%CI -17.97 to -15.89, $p$<0.00001, $I^2$ = 96%), and the $LSpO_2$ increased by 7.77% (95%CI 7.02% to 8.52%, $p$<0.00001, $I^2$ = 94%). The RDI synthesized with two studies showed a significant decrease of -5.92 events/h (95%CI -6.65 to -5.19, $p$<0.00001, $I^2$ = 0%).

Only a few studies reported treatment outcomes of sleep structures (Fig 5A and 5B). With MAD fit, sleep efficiency showed an insignificant change of 1.05% (95%CI -1.93 to 4.02%, $p$ = 0.49, $I^2$ = 77%). The arousal index decreased by -15.26/h (95%CI -18.97 to -11.55, $p$<0.00001, $I^2$ = 54%).

As for the treatment efficacy on comorbidities (Fig 6A to 6C), long-term use of MAD could lower systolic blood pressure by -4.31 mmHg (95%CI -6.31 to -2.31, $p$<0.0001, $I^2$ = 0%) and diastolic blood pressure by -7.75 mmHg (95%CI -14.57 to -0.93, $p$ = 0.03, $I^2$ = 59%). Functional abilities to perform daily activities measured with the Functional outcomes of sleep questionnaire (FOSQ) increased with long-term use of MAD by 3.16 (95%CI 2.62 to 3.70, $p$<0.00001, $I^2$ = 6%).

The synthesized treatment compliance between MAD and CPAP is shown in Fig 7. Dropout rates between these two treatment modalities were insignificant (OR = 0.88, 95%CI 0.52 to 1.47, $p$ = 0.62, $I^2$ = 44%).

## Risk of publication bias

The funnel plot of AHI in the treatment duration subgroup is illustrated in Fig 8. Studies with treatment duration >5y were fewer than those with treatment duration <5y. The distribution of studies was overall symmetrical.

## Discussion

In this study, we conducted a systematic review and meta-analysis to determine the efficacy of the long-term use of MAD for the treatment of adult OSA. A total of 22 studies were included

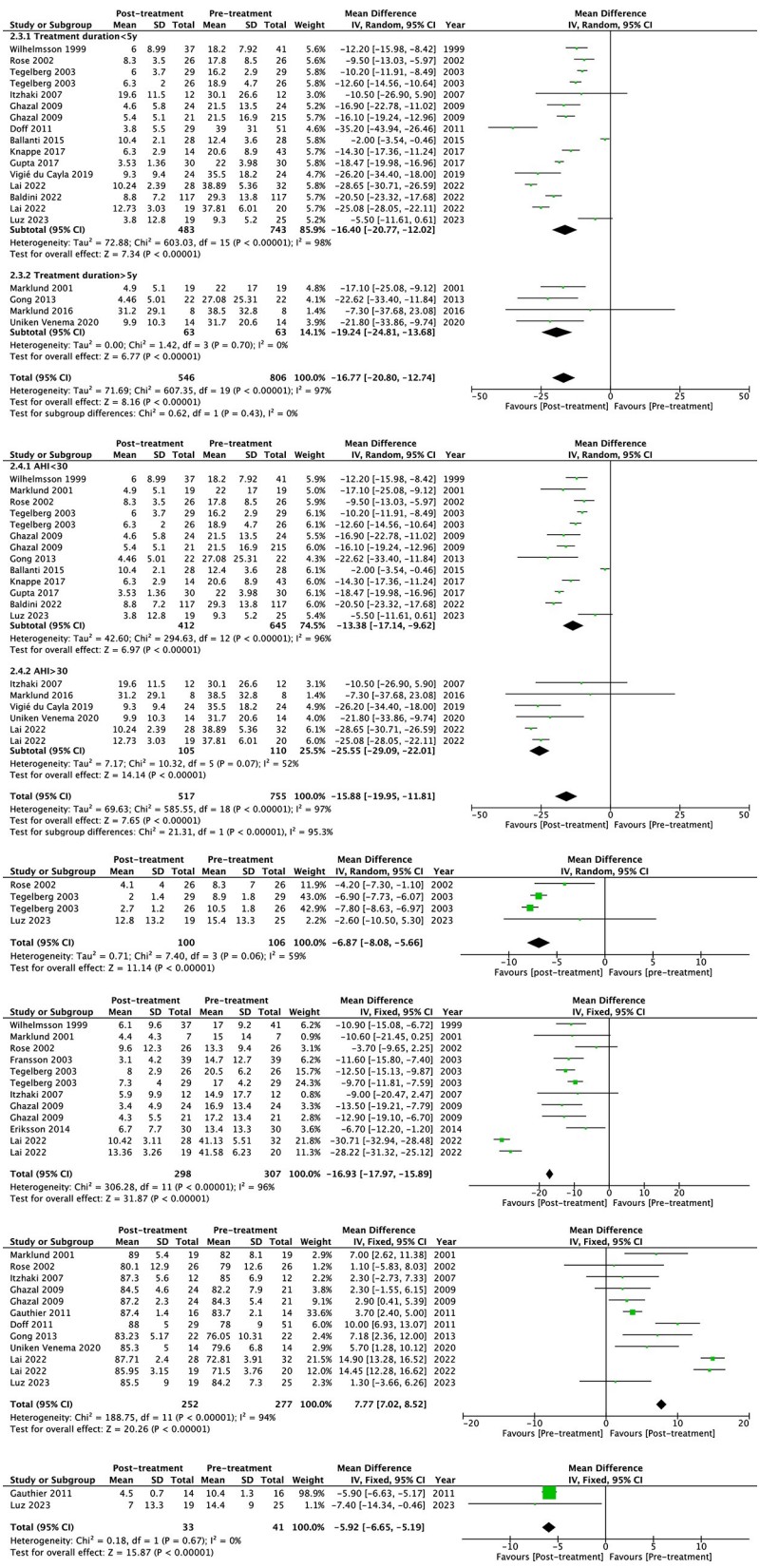

**Fig 4. The forest plot of the mean difference of objective efficacy of mandibular advancement devices (MAD) on obstructive sleep apnea (OSA).** (a) Apnea hypopnea index (AHI), sub-grouped by treatment duration, (b) Apnea hypopnea index (AHI), sub-grouped by baseline severity, (c) Apnea index (AI), (d) Oxygen desaturation index (ODI), (e) The lowest oxygen saturation (LSpO$_2$), and (f) Respiratory disturbance index (RDI).

in the systematic review, and the meta-analysis was conducted with 20 studies. The results of the study suggested that the long-term effect of MAD in the treatment of OSA is reliable. Subjective efficacy, represented by ESS, and objective parameters, such as AHI, ODI, and LSpO$_2$, both showed significant improvements with MAD. Limited evidence also suggested long-term use of MAD could lower blood pressure and increase daily functional activities.

This study found that long-term wearing of MAD could improve AHI, which was not affected by the baseline severity of OSA. The effect of MAD on subjective daytime sleepiness measured using the ESS or PSQI followed a similar pattern but these instruments were less sensitive to differences than AHI [49]. AHI, derived from polysomnography or HSAT, is the most commonly used objective parameter to evaluate the treatment efficacy. The subgroup analysis of the treatment duration of less than and greater than 5 years also suggested that the improvement of MAD on AHI has long-term stability. However, there was high heterogeneity across studies. Patients who suffer from severe OSA or require immediate treatment due to comorbidities have often been excluded from the use of MAD as a therapeutic option. Furthermore, patients who completed long-term follow-up might exhibit a higher likelihood of positive response to the treatment, which might overestimate the efficacy of MAD. CPAP is generally more effective than MAD in improving respiratory events. The pooled results of the current study need to be verified with future studies with larger sample sizes and longer follow-up periods.

The therapeutic effect of MAD depends on patients' compliance. This study did not find the dropout rates between MAD and CPAP to be significantly different. There is some evidence of better compliance and patient preference in favor of MAD as compared to CPAP, with the average wearing time of MAD 1.1 hours longer per night [39]. Objectively measured compliance with a microsensor thermometer during MAD treatment suggested good compliance, with 83% of patients using MADs for >4h/night at 1 year follow-up [50]. A meta-analysis

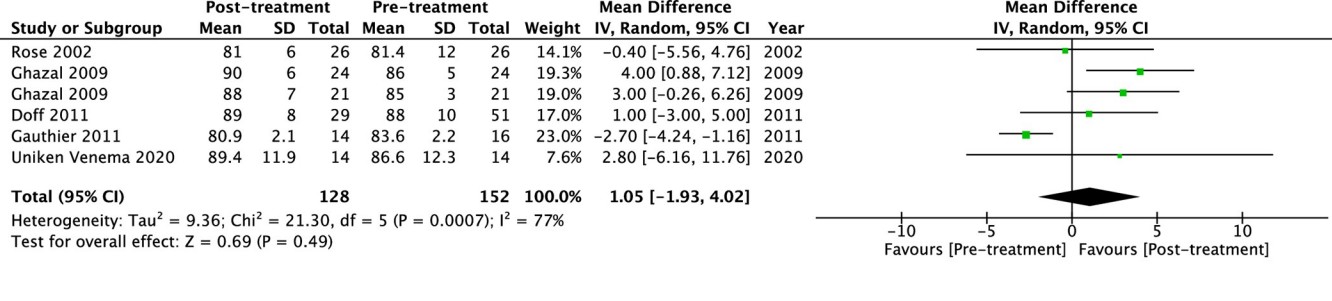

**Fig 5. The forest plot of the mean difference of sleep structures of mandibular advancement devices (MAD) on obstructive sleep apnea (OSA).** (a) Sleep efficiency and (b) Arousal index (ArI).

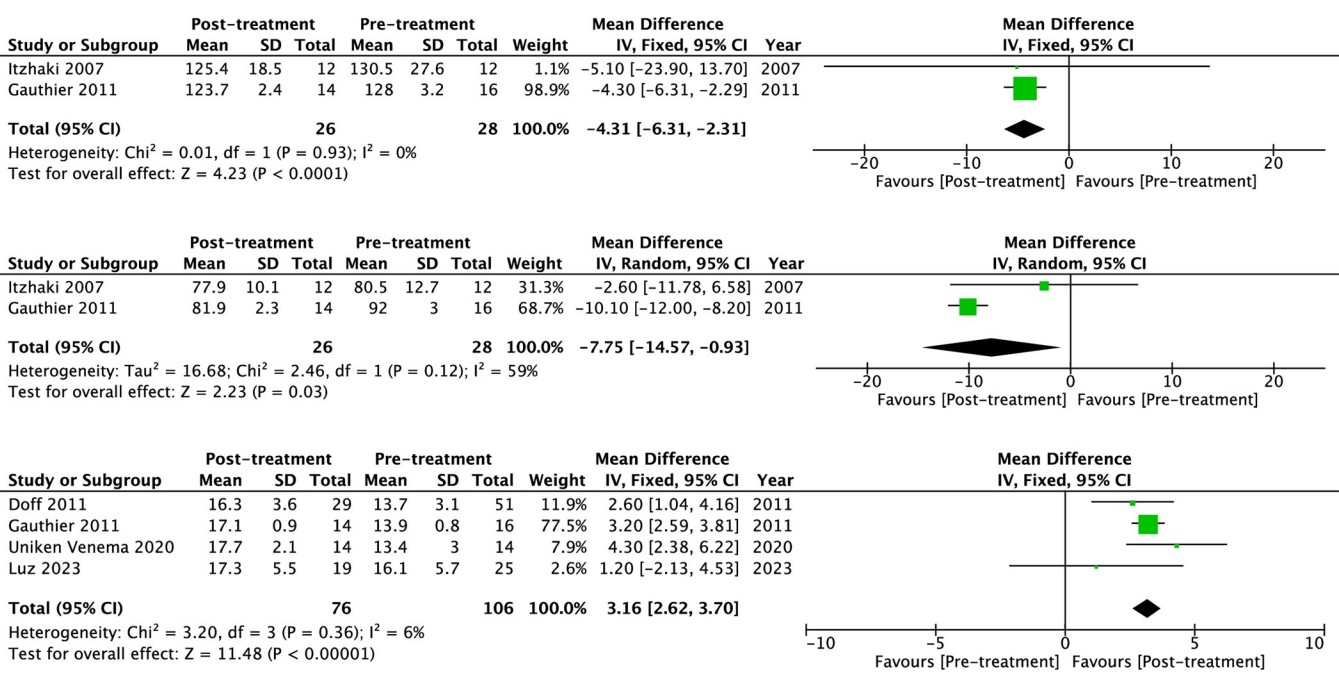

**Fig 6. The forest plot of the mean difference of blood pressure and functional abilities of mandibular advancement devices (MAD) on obstructive sleep apnea (OSA).** (a) Systolic blood pressure, (b) Diastolic blood pressure, and (c) Functional outcomes of sleep questionnaire (FOSQ).

found that discontinuing therapy from side effects was significantly lower due to the use of MADs than CPAP [51]. Although there is some evidence that MAD may be better tolerated than CPAP in the short term, the limited long-term data available suggest that adherence to this less invasive intervention also decreases over time [49]. For patients with long-term MAD treatment of OSA, regular follow-up is required. Short-term follow-up (generally 3 to 6 months) could relieve patients' discomfort in an early stage to improve compliance. At the same time, the objective efficacy of MAD could be evaluated to adjust the protrusion of MAD accordingly. Long-term follow-up is generally recommended to be conducted once a year [6], to examine the efficacy of MAD, patient's compliance, dental health status, occlusion, and temporomandibular joint.

This study found limited evidence that long-term use of MAD could improve blood pressure and functional abilities. Previous studies found that there is good evidence that CPAP improves daytime sleepiness, cognitive function, quality of life, and cardiovascular risk factors

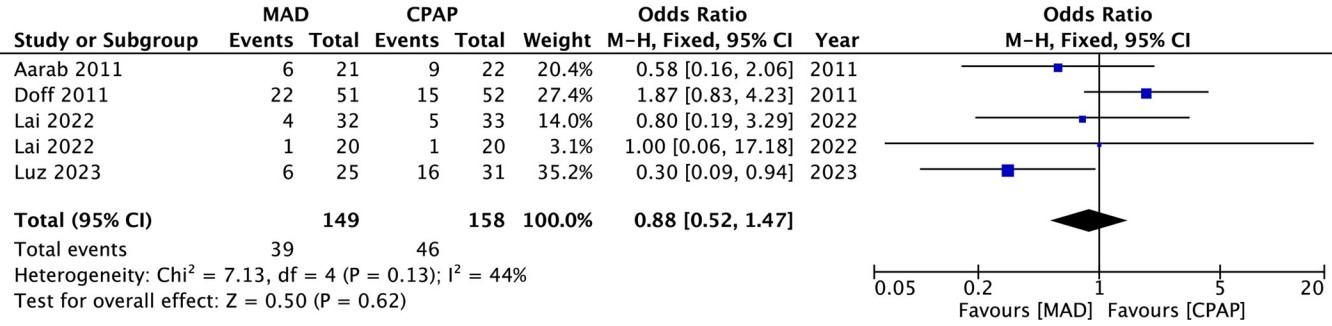

**Fig 7. The forest plot of the dropout rates between mandibular advancement devices (MAD) and continuous positive airway pressure (CPAP).**

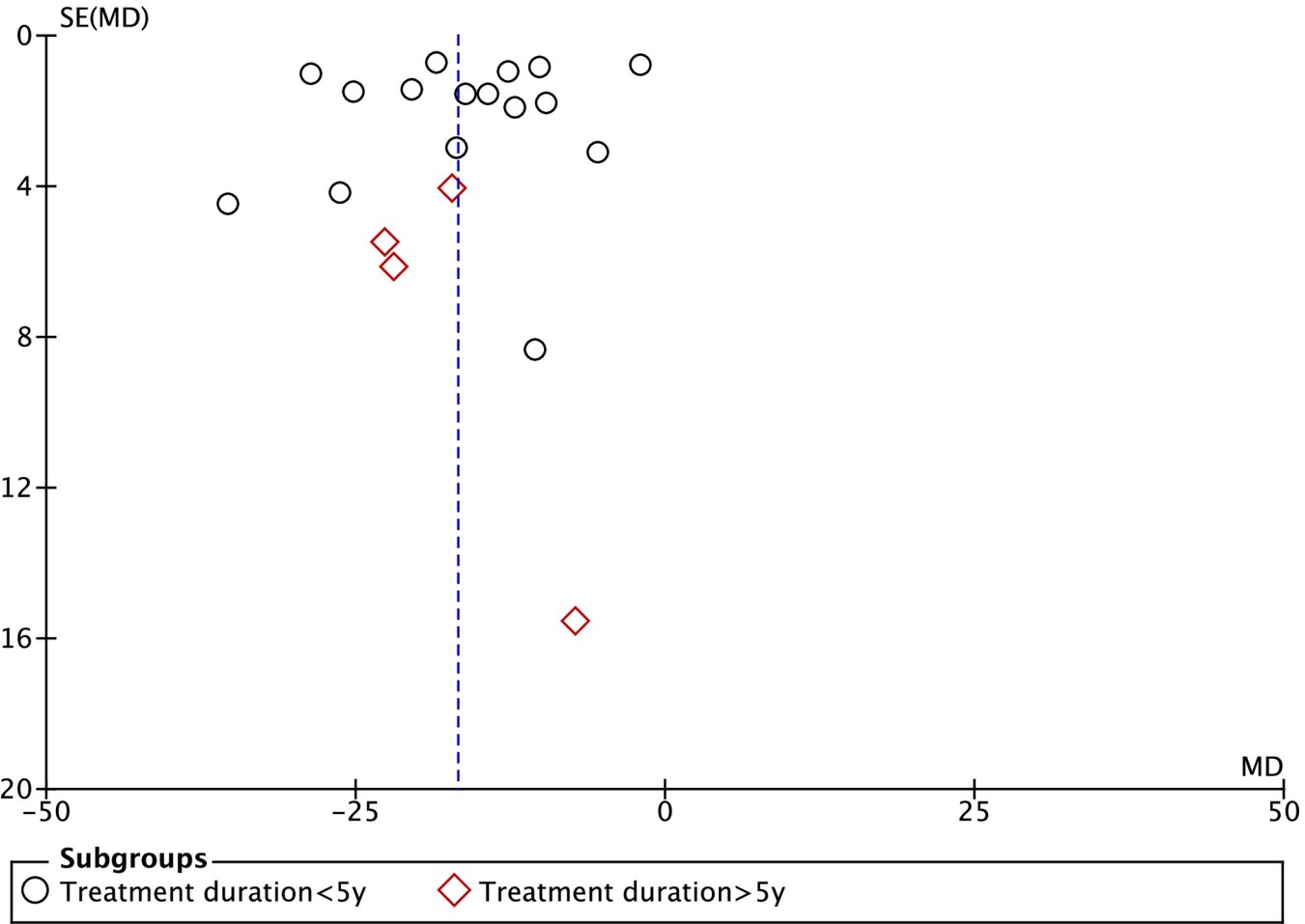

**Fig 8. The funnel plot for the data of apnea-hypopnea index (AHI).**

of patients with OSA [49]. Some studies found that the improvement of subjective and objective sleepiness, quality of life, and cognitive function produced by MADs was not inferior to that reported with CPAP therapy in mild to moderate OSA [52, 53], whereas CPAP is more effective in severe cases [49]. Systolic and diastolic 24-hour blood pressure were similar under CPAP and MAD treatments as well [53, 54]. However, the effects of MADs on comorbidities, quality of life, and road traffic accident risk were unable to be synthesized due to the small number of studies and inconsistent methodology [55]. Larger and longer RCTs examining the benefits of MAD treatment to cardiac, metabolic, neurocognitive healthcare, and medication use are needed in the future. Treatment modalities should be re-evaluated for MAD-treated patients when they develop recurrent symptoms, show substantial weight changes, or receive diagnoses of comorbidities relevant to OSA [51].

Beneficial treatment effects may be reduced by treatment-related side effects, and most side effects of MADs are dental-related [22], including the discomfort of teeth and temporomandibular joint, sore orofacial muscle in short-term use, and occlusal changes in anterior teeth, which increased with treatment duration. A custom-made and titratable appliance was recommended over other types of appliances by the AASM [51], to achieve treatment efficacy in a minimal mandibular protrusion, increasing comfort and efficacy and decreasing possible side

effects [28]. With recently published studies with a follow-up period of more than 10 years [31, 36, 37], it might be beneficial to review the side effects of long-term MAD use.

There were several limitations to this study. First, due to ethical considerations, the study design for the long-term efficacy of MAD is mainly self-controlled trials, and the quality of the studies is medium. Second, the follow-up time of the original studies was long, and the drop-out rate was high, resulting in a certain degree of bias in the results. Third, in almost all comparisons, there was significant heterogeneity across studies. Although some of this could be explained by baseline severity, design, and treatment duration, unexplained heterogeneity remains. Although this study used random-effects meta-analysis to provide unbiased and robust estimates, further elucidation of the sources of heterogeneity would be useful.

## Conclusion

Moderate evidence suggests that the subjective and objective effect of MAD on adult OSA has long-term stability. Limited evidence suggests long-term use of MAD might improve comorbidities and healthcare. In clinical practice, regular follow-up is recommended.

## Supporting information

**S1 Checklist. PRISMA 2009 checklist.**
(DOC)

## Acknowledgments

The authors thank all investigators and supporters involved in the study.

## Author Contributions

**Conceptualization:** Min Yu, Xuemei Gao.

**Data curation:** Min Yu, Yanyan Ma.

**Funding acquisition:** Xuemei Gao.

**Investigation:** Min Yu.

**Methodology:** Min Yu, Yanyan Ma.

**Project administration:** Min Yu.

**Software:** Min Yu.

**Writing – original draft:** Min Yu, Yanyan Ma.

**Writing – review & editing:** Fang Han, Xuemei Gao.

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
