## [Decision Letter · Decision Letter 0]

22 Aug 2023

PONE-D-23-24787Long-term efficacy of mandibular advancement devices in the treatment of adult obstructive sleep apnea: a systematic review and meta-analysisPLOS ONE

Dear Dr. Gao,

Thank you for submitting your manuscript to PLOS ONE. After careful consideration, we feel that it has merit but does not fully meet PLOS ONE’s publication criteria as it currently stands. Therefore, we invite you to submit a revised version of the manuscript that addresses the points raised during the review process.

We look forward to receiving your revised manuscript.

Kind regards,

Ji Woon Park

Academic Editor

PLOS ONE

Journal Requirements:

"This study was supported by the Clinical Research Foundation of Peking University School and Hospital of Stomatology (PKUSS-2023CRF106) granted to M.Y."

4. "PLOS requires an ORCID iD for the corresponding author in Editorial Manager on papers submitted after December 6th, 2016. Please ensure that you have an ORCID iD and that it is validated in Editorial Manager. To do this, go to ‘Update my Information’ (in the upper left-hand corner of the main menu), and click on the Fetch/Validate link next to the ORCID field. This will take you to the ORCID site and allow you to create a new iD or authenticate a pre-existing iD in Editorial Manager. Please see the following video for instructions on linking an ORCID iD to your Editorial Manager account: https://www.youtube.com/watch?v=_xcclfuvtxQ

Reviewers' comments:

Reviewer's Responses to Questions

**Comments to the Author**

1. Is the manuscript technically sound, and do the data support the conclusions?

Reviewer #1: Yes

Reviewer #2: Yes

2. Has the statistical analysis been performed appropriately and rigorously? 

Reviewer #1: Yes

Reviewer #2: Yes

3. Have the authors made all data underlying the findings in their manuscript fully available?

Reviewer #1: No

Reviewer #2: No

4. Is the manuscript presented in an intelligible fashion and written in standard English?

Reviewer #1: Yes

Reviewer #2: Yes

5. Review Comments to the Author

Reviewer #1: The paper is a classical technical Meta Analyses of breathing sleep related findings on efficacy of MAD; oral appliance used to manage sleep apnea.

It is of interests although it is confirmatory in nature. The presentation of most relevant outcome, hypoxia, should be added to the Abstract. It is a little odd, the most relevant outcomes for health and survival plus debate on compliance (CPAP over MAD) are eluded…. In the actual format, the paper is outdated; please be in line with actual debate and literature.

Missing information (probably not available in papers revised) on comorbidities, medications used are missing. If not available add to Discussion for suggestions of future studies.

Similarly, most studies using MAD did not assess compliance objectively, new technological development allow such estimation. It should be also add to Discussion.

Plus, some controversy on risk of side effect, add to Discussion. This was in Introduction but absent of Discussion.

Reviewer #2: This is a very much needed manuscript, because the long-term outcome of MAD therapy in patients with OSA is little known. There are few studies available and no comprehensive reviews or meta-analyses of these important outcomes. Since patients discontinue treatment or do not want to participate in follow-up studies, there are also large drop-out rates in the longer term studies. The study is easy to read. But, there are, however, some concerns about the reporting and analysis of the findings, some missing studies and more discussion about the meaning of the high drop-out rates for the total interpretation of the results. .

1. Table 1. The authors describe a mixture of values regarding the number of patients in the original samples and the analysed number of patients, which makes column 4 (N) and column 10 (Dropouts) difficult to understand and compare. All these data have to be controlled and reported in a logic manner. The numbers reported have to be better explained in the Table, since it becomes impossible to understand that, for example the Baldini et al. retrospective study (reference 46) describes N=117 and Dropouts as 327 patients. It was actually, originally 444 patients, 327 dropped out or denied participation and the remaining 117 patients were evaluated. In the RCT studies, for instance Tegelberg et al. (reference 28) these figures means that 74 patients were originally included, 19 dropped out, leaving 55 for analysis. The Table must be designed so that the data become understandable. The authors also need to include, not only patients who actually dropped out, but also those who were excluded of other reasons, for instance the Rose et al. study (reference 38) included 86 patients, only 26 were evaluated, but the authors only describe 23 patients as drop-outs. These 23 patients constituted only a subgroup of a subgroup. The data should be described in 4 columns according to:

Original sample - Dropouts (N +%) - Analysed (N, %) - Women/Men (%) in analysed sample

2. The authors should comment something about the exclusion criteria in these included studies in the discussion, since the studies do not include patients with for instance; an urgent need for OSA-treatment because of comorbidities or dental problems. This influences comparisons with results from long term outcomes for PAP, in addition a much more effective treatment in terms of AHI reductions.

3. The authors should include more about the influence of the high drop-out rates for the interpretation of the present findings as well as include something about that in the conclusion. It is not enough, to just mention this in the limitations.

4. On page 12, lines 17-19, the authors write: “However, previous studies found that there was no difference between MAD and PAP in improving subjective and objective sleepiness, quality of life, cognitive function, and blood pressure.” This statement probably only applies to mild to moderate cases regarding blood pressure (Randerath, W., Verbraecken, J., et al. 2021) and there also some results that PAP is more effective on daytime sleepiness in more severe cases (Sharples, L.D., Clutterbuck-James, A.L., et al. 2016). Please modify this statement, also with respect to point 3 above, where it must be mentioned that many patients are excluded in the MAD studies.

5. There are more longer term studies available that should be included, if they meet the quality criteria for the present review (Doff, M.H., Hoekema, A., et al. 2013, eSilva, L.O., Guimaraes, T.M., et al. 2021, Gauthier, L., Laberge, L., et al. 2011, Haviv, Y., Bachar, G., et al. 2015)

Doff MH, Hoekema A, Wijkstra PJ, van der Hoeven JH, Huddleston Slater JJ, de Bont LG, Stegenga B (2013) Oral appliance versus continuous positive airway pressure in obstructive sleep apnea syndrome: a 2-year follow-up. Sleep 36: 1289-1296

eSilva LO, Guimaraes TM, Pontes G, Coelho G, Badke L, Fabbro CD, Tufik S, Bittencourt L, Togeiro S (2021) The effects of continuous positive airway pressure and mandibular advancement therapy on metabolic outcomes of patients with mild obstructive sleep apnea: a randomized controlled study. Sleep Breath 25: 797-805

Gauthier L, Laberge L, Beaudry M, Laforte M, Rompre PH, Lavigne GJ (2011) Mandibular advancement appliances remain effective in lowering respiratory disturbance index for 2.5-4.5 years. Sleep Medicine 12: 844-849

Haviv Y, Bachar G, Aframian DJ, Almoznino G, Michaeli E, Benoliel R (2015) A 2-year mean follow-up of oral appliance therapy for severe obstructive sleep apnea: a cohort study. Oral Diseases 21: 386-392

Randerath W, Verbraecken J, de Raaff CAL, Hedner J, Herkenrath S, Hohenhorst W, Jakob T, Marrone O, Marklund M, McNicholas WT, et al. (2021) European Respiratory Society guideline on non-CPAP therapies for obstructive sleep apnoea. European Respiratory Review: An Official Journal of the European Respiratory Society 30

Sharples LD, Clutterbuck-James AL, Glover MJ, Bennett MS, Chadwick R, Pittman MA, Quinnell TG (2016) Meta-analysis of randomised controlled trials of oral mandibular advancement devices and continuous positive airway pressure for obstructive sleep apnoea-hypopnoea. Sleep Medicine Reviews 27: 108-124

6. PLOS authors have the option to publish the peer review history of their article (what does this mean?). If published, this will include your full peer review and any attached files.

Reviewer #1: **Yes: **GL

Reviewer #2: No

---

## [Author Response · Author response to Decision Letter 0]

12 Sep 2023

Dear Editor and Reviewers,

Thank you for your comments concerning our manuscript entitled “Long-term efficacy of mandibular advancement devices in the treatment of adult obstructive sleep apnea: a systematic review and meta-analysis” (PONE-D-23-24787). We greatly appreciate all the comments provided on our paper. We have thoroughly reviewed each comment and made revisions accordingly with the hope of meeting approval. The revised sections have been clearly marked in the article. The main correction and response to the reviewers' comments are outlined below:

Responds to the reviewers’ comments:

Reviewer 1:

1. The paper is a classical technical Meta Analyses of breathing sleep related findings on efficacy of MAD; oral appliance used to manage sleep apnea.

It is of interests although it is confirmatory in nature. The presentation of most relevant outcome, hypoxia, should be added to the Abstract. It is a little odd, the most relevant outcomes for health and survival plus debate on compliance (CPAP over MAD) are eluded…. In the actual format, the paper is outdated; please be in line with actual debate and literature.

Respond: Thank you so much for sharing your valuable suggestions with us. We are truly grateful for your input and have taken your feedback to heart. We have made some important changes to the article based on your suggestions, including incorporating synthesized outcomes related to RDI, sleep efficiency, arousal index, blood pressure, and functional abilities as represented by FOSQ in the meta-analysis. We have also included comparisons of dropout rates between MAD and CPAP based on four RCTs to address the issue of compliance. These changes will go a long way in improving the readability and overall quality of the article. Thank you again for your help in this matter. Please find the revised content on Page 12.

2. Missing information (probably not available in papers revised) on comorbidities, medications used are missing. If not available add to Discussion for suggestions of future studies.

Respond: Thank you for providing your valuable feedback. Most articles did not adequately address outcomes related to comorbidities and medication use, as you predicted. Two articles reported the change in blood pressure after long-term treatment with MAD, and we have added the outcome to the manuscript (Page 13, Line 3). As per your comment, we have included recommendations for further research in the discussion section. Please find the revised content on Page 16.

3. Similarly, most studies using MAD did not assess compliance objectively, new technological development allow such estimation. It should be also add to Discussion.

Plus, some controversy on risk of side effect, add to Discussion. This was in Introduction but absent of Discussion.

Respond: We truly appreciate your suggestions. The addition of objective compliance and side effects of MAD use to the discussion has certainly made it more informative and valuable. Thank you for your contribution. Please find the revised content on Page 15.

Thank you for taking the time to share your thoughts with us. Your input is valuable and we sincerely appreciate it.

Reviewer 2:

1. Table 1. The authors describe a mixture of values regarding the number of patients in the original samples and the analysed number of patients, which makes column 4 (N) and column 10 (Dropouts) difficult to understand and compare. All these data have to be controlled and reported in a logic manner. The numbers reported have to be better explained in the Table, since it becomes impossible to understand that, for example the Baldini et al. retrospective study (reference 46) describes N=117 and Dropouts as 327 patients. It was actually, originally 444 patients, 327 dropped out or denied participation and the remaining 117 patients were evaluated. In the RCT studies, for instance Tegelberg et al. (reference 28) these figures means that 74 patients were originally included, 19 dropped out, leaving 55 for analysis. The Table must be designed so that the data become understandable. The authors also need to include, not only patients who actually dropped out, but also those who were excluded of other reasons, for instance the Rose et al. study (reference 38) included 86 patients, only 26 were evaluated, but the authors only describe 23 patients as drop-outs. These 23 patients constituted only a subgroup of a subgroup. The data should be described in 4 columns according to:

Original sample - Dropouts (N +%) - Analysed (N, %) - Women/Men (%) in analysed sample.

Respond: Thank you so much for your suggestions. We have adjusted the table based on your advice to improve readability. Please find the revised content on Pages 7 to 10.

2. The authors should comment something about the exclusion criteria in these included studies in the discussion, since the studies do not include patients with for instance; an urgent need for OSA-treatment because of comorbidities or dental problems. This influences comparisons with results from long term outcomes for PAP, in addition a much more effective treatment in terms of AHI reductions.

Respond: We appreciate your suggestion and understand the importance of considering exclusion criteria when estimating MAD efficacy. Thank you for bringing this to our attention. Please find the revised content on Page 15.

3. The authors should include more about the influence of the high drop-out rates for the interpretation of the present findings as well as include something about that in the conclusion. It is not enough, to just mention this in the limitations.

Respond: We appreciate your suggestions. It was a productive discussion where we highlighted the impact of high dropout rates on the overestimation of MAD treatment. It's important to consider all factors that can affect the efficacy of the treatment, as you commented. Please find the revised content on Page 15.

4. On page 12, lines 17-19, the authors write: “However, previous studies found that there was no difference between MAD and PAP in improving subjective and objective sleepiness, quality of life, cognitive function, and blood pressure.” This statement probably only applies to mild to moderate cases regarding blood pressure (Randerath, W., Verbraecken, J., et al. 2021) and there also some results that PAP is more effective on daytime sleepiness in more severe cases (Sharples, L.D., Clutterbuck-James, A.L., et al. 2016). Please modify this statement, also with respect to point 3 above, where it must be mentioned that many patients are excluded in the MAD studies.

Respond: We appreciate your valuable input. We have taken note of your suggestion and updated our information to provide a clearer understanding of how MAD can effectively address comorbidities depending on the severity of OSA, as well as its long-term benefits. Thank you for helping us improve our content. Please find the revised content on Page 16.

5. There are more longer term studies available that should be included, if they meet the quality criteria for the present review (Doff, M.H., Hoekema, A., et al. 2013, eSilva, L.O., Guimaraes, T.M., et al. 2021, Gauthier, L., Laberge, L., et al. 2011, Haviv, Y., Bachar, G., et al. 2015)

Respond: Thank you for the comment. The studies conducted by Doff et al and Gauthier et al were added to the included studies. We sincerely apologize for our mistake.

The study conducted by Haviv et al in 2015 (doi: 10.1111/odi.12291) was excluded because the follow-up period was “24 months (range, 5–76 months)”, indicating some participants were followed less than a year. The study conducted by e Silva et al in 2021 (doi: 10.1007/s11325-020-02183-1) and that conducted by Luz et al in 2023 (doi: 10.1007/s11325-022-02694-z) had the same study design and participants, and these two studies shared the same results of before and after-MAD treatment PSG parameters. Therefore, we only included the latest manuscript of Luz’s study.

Other studies are high-quality review articles, which were excluded from the meta-analysis but were added to the discussion where appropriate. Your suggestions have been greatly appreciated. 

We’re grateful for your insightful input. Thank you so much for sharing your thoughts with us!

---

## [Decision Letter · Decision Letter 1]

2 Oct 2023

Long-term efficacy of mandibular advancement devices in the treatment of adult obstructive sleep apnea: a systematic review and meta-analysis

PONE-D-23-24787R1

Dear Dr. Gao,

We’re pleased to inform you that your manuscript has been judged scientifically suitable for publication and will be formally accepted for publication once it meets all outstanding technical requirements.

Kind regards,

Ji Woon Park

Academic Editor

PLOS ONE

**Comments to the Author**

1. Reviewer #1: All comments have been addressed

Reviewer #2: All comments have been addressed

2. Is the manuscript technically sound, and do the data support the conclusions?

Reviewer #1: Yes

Reviewer #2: Yes

3. Has the statistical analysis been performed appropriately and rigorously? 

Reviewer #1: Yes

Reviewer #2: Yes

4. Have the authors made all data underlying the findings in their manuscript fully available?

Reviewer #1: Yes

Reviewer #2: Yes

5. Is the manuscript presented in an intelligible fashion and written in standard English?

Reviewer #1: Yes

Reviewer #2: Yes

6. Review Comments to the Author

Reviewer #1: (No Response)

Reviewer #2: All my comments have been excellently considered and the manuscript is much easier to read, in particular the Table. The authors have also included more available studies, leading to a very updated review.

Reviewer #1: **Yes: **GJ Lavigne

Reviewer #2: No

---

## [Editor Report · Acceptance letter]

5 Oct 2023

PONE-D-23-24787R1 

Long-term efficacy of mandibular advancement devices in the treatment of adult obstructive sleep apnea: a systematic review and meta-analysis 

Dear Dr. Gao:

I'm pleased to inform you that your manuscript has been deemed suitable for publication in PLOS ONE. Congratulations! Your manuscript is now with our production department. 

Kind regards, 

on behalf of

Professor Ji Woon Park 

Academic Editor

PLOS ONE